# Graph Theory-Based Electroencephalographic Connectivity and Its Association with Ketogenic Diet Effectiveness in Epileptic Children

**DOI:** 10.3390/nu13072186

**Published:** 2021-06-25

**Authors:** Ting-Yu Su, Pi-Lien Hung, Chien Chen, Ying-Jui Lin, Syu-Jyun Peng

**Affiliations:** 1Division of Pediatric Neurology, Department of Pediatrics, Kaohsiung Chang Gung Memorial Hospital and Chang Gung University College of Medicine, Kaohsiung 83301, Taiwan; b101097123@cgmh.org.tw (T.-Y.S.); flora1402@cgmh.org.tw (P.-L.H.); 2Department of Neurology, Neurological Institute, Taipei Veterans General Hospital and School of Medicine, National Yang Ming Chiao Tung University College of Medicine, Taipei 11217, Taiwan; cchien0604@gmail.com; 3Division of Pediatric Cardiology, Department of Pediatrics, Kaohsiung Chang Gung Memorial Hospital and Chang Gung University College of Medicine, Kaohsiung 83301, Taiwan; rayray@cgmh.org.tw; 4Professional Master Program in Artificial Intelligence in Medicine, College of Medicine, Taipei Medical University, 19F, No.172-1, Sec. 2, Keelung Rd., Da’an Dist., Taipei City 10675, Taiwan

**Keywords:** ketogenic diet therapy, drug-resistant epilepsy, interictal electroencephalography, graph theory, functional connectivity

## Abstract

Ketogenic diet therapies (KDTs) are widely used treatments for epilepsy, but the factors influencing their responsiveness remain unknown. This study aimed to explore the predictors or associated factors for KDTs effectiveness by evaluating the subtle changes in brain functional connectivity (FC) before and after KDTs. Segments of interictal sleep electroencephalography (EEG) were acquired before and after six months of KDTs. Analyses of FC were based on network-based statistics and graph theory, with a focus on different frequency bands. Seventeen responders and 14 non-responders were enrolled. After six months of KDTs, the responders exhibited a significant functional connectivity strength decrease compared with the non-responders; reductions in global efficiency, clustering coefficient, and nodal strength in the beta frequency band for a consecutive range of weighted proportional thresholds were observed in the responders. The alteration of betweenness centrality was significantly and positively correlated with seizure reduction rate in alpha, beta, and theta frequency bands in weighted adjacency matrices with densities of 90%. We conclude that KDTs tended to modify minor-to-moderate-intensity brain connections; the reduction of global connectivity and the increment of betweenness centrality after six months of KDTs were associated with better KD effectiveness.

## 1. Introduction

Ketogenic dietary therapies (KDTs) are non-pharmacologic treatments based on diets with high fat, low carbohydrate, and adequate protein content. They are widely used in the field of refractory epilepsy and neurometabolic disorders. Nowadays, several dietary protocols have been established and confirmed to be effective, such as the classic ketogenic diet (cKD), the medium-chain triglyceride diet, and the modified Atkins diet. A systemic review indicated that the use of KDTs in childhood epilepsy produced a reduction of more than 50% in seizure frequency in 33% of patients, and 15.6% of patients were seizure-free [1]. Furthermore, they are the standard therapies for patients with glucose transporter 1 deficiency syndrome and pyruvate dehydrogenase deficiency syndrome [2].

However, every patient responds to KDTs differently, and few factors have been identified that convincingly predict its effectiveness. In previous studies, age, gender, intellectual status, and common variants in established candidate genes, such as *KCNJ11* and *BAD*, did not predict the response to KDTs [3,4,5,6], nor did initiation protocol type or blood glucose concentration during treatment [7]. Children with generalized tonic–clonic seizures seemed to experience better outcomes than those without [8,9], and those with complex partial seizures as the main seizure type were less likely to have a dramatic response to treatment [10]. After the initiation of KDTs, patients with ≥50% seizure reduction at three months were more likely to have a good response to treatment after 12 months [5].

In recent years, research has focused on electroencephalography (EEG) findings. A study showed that after three months of KDTs, the number of interictal epileptiform discharges (IEDs) was reduced significantly, especially during sleep [11]. The presence of IED in the temporal area has been associated with a poor response to the treatment [12]. Those with a ≥10% decrease in the frequency of IED at one month were more likely to respond well to the treatment [13]. At six weeks, a proportional reduction of 30% in the IED index in sleep EEG was associated with being responsive to the treatment [14]. To take a deeper look at the EEG change, we conducted this study to investigate the predictors or the associated factors for KD effectiveness based on graph theory to evaluate the serial changes in brain functional connectivity before and six months after KDTs. This study hypothesized that the changes in graph-theoretical brain functional connectivity could serve as parameters for evaluating KD effectiveness after receiving it for six months.

## 2. Materials and Methods

### 2.1. Participants

From January 2018 to January 2019, children in Kaohsiung Chang Gung Memorial Hospital with drug-resistant epilepsy were enrolled and underwent a KDT program. This study was approved by the Institutional Review Board of Chang Gung Memorial Hospital (201700968A3). Informed consent was signed by the subjects’ parents or guardians before enrollment. Drug-resistant epilepsy was defined as “the failure to achieve sustained seizure freedom with adequate uses of two antiepileptic drugs (AEDs), either as monotherapy or in combination” [15]. Children diagnosed with inborn errors of metabolism were excluded, as were those who were candidates for epilepsy surgery.

### 2.2. Study Protocol

We arranged for all study subjects to be admitted to the pediatric ward of Kaohsiung Chang Gung Memorial Hospital. Screening for contraindications to KDTs, which included the measurement of serum ammonia, lactate, cholesterol, triglyceride, amino acids and urinary organic acids and the family history of porphyria was performed before treatment initiation.

After screening, patients received a non-fasting gradual KD initiation protocol (GRAD-KD), beginning on a five-day classic ketogenic diet (cKD) in the pediatric ward. The energy requirement of each patient was calculated by a registered dietitian based on age, body weight, body height, daily activity, and dietary history of each subject. Therapy was initiated without fasting, as follows: patients received 1/9 of the Recommended Dietary Allowance (RDA) calories on the first day, 1/6 on day 2, 1/3 on day 3, 2/3 on day 4, and achieved full RDA calories on day 5. Ketogenic ratios used were 2:1 on day 1 and were elevated slowly to 3–4:1 on day 5. Daily protein intake ranged from 1.5 to 2.5 g/kg, based on the subjects’ age and physical status. We added medium-chain triglyceride powder gradually to the diet to achieve a concentration of at least 40 g/day. During the program, plasma sugar and beta-hydroxybutyrate (βHB) levels were measured every 2 to 4 hours to avoid the events of hypoglycemia and hyperketosis. The frequency of seizure was recorded in a seizure diary by guardians, and the βHB levels were measured during monthly outpatient visits. All subjects were followed up for at least six months. A responder was defined as a subject with a ≥50% reduction in seizure frequency.

### 2.3. EEG Recording

EEG recordings were acquired in 31 children during sleep. EEG were sampled at 125 Hz. EEG activity was recorded using 19 electrodes (Nicolette V32) referenced to the Cz electrode and positioned according to the 10–20 international system of electrode placement.

### 2.4. EEG Preprocessing

Continuous scalp EEG data were imported into EEGLAB v2019.0, a MATLAB-based open toolbox [16]. A segment that consisted of a one-min recording during sleep stage I–II was selected. Each EEG channel was band-pass filtered using a finite impulse response filter in the 0.5–30 Hz frequency band. Data were re-referenced to the average of all scalp channels. Subsequently, an independent component analysis (ICA) decomposition was performed [17] to remove eye movements, blinks, and other mechanical artifacts from EEG. The EEG were then partitioned into 29 epochs with a duration of 4 s and an overlap of 2 s. The epochs were examined to guarantee that none of them involved bad channels and none of them contained head motion or muscle movement. To explore the functional network across broad brain regions, functional connection strength was estimated with phase locking value (PLV) across all electrodes (Fp1, Fp2, F3, F4, C3, C4, P3, P4, O1, O2, F7, F8, T3, T4, T5, T6, Fz, Cz, and Pz) for delta (0.5–4 Hz), theta (4–8 Hz), alpha (8–13 Hz), and beta (13–30 Hz) frequency bands.

### 2.5. Functional Connectivity

We used the PLV to reconstruct functional networks between all pairs of 19 electrodes for each region of frequency band and epoch, separately. In order to reduce the effect of volume conduction, we selected a PLV measure for phase synchronization quantitation [18]. The PLV is a measurement of the asymmetry in the distribution of instantaneous phase differences [18]. A distribution which is centered around zero and symmetric may represent spurious connectivity, and a flat distribution represents no connectivity. Deviations from a symmetric distribution represent dependency between sources. The PLV ranges between 0 and 1. A value of “zero” indicates that no coupling occurs, and a value of “one” indicates perfect phase locking. For each connectivity measure, weighted adjacency matrices were produced by separately averaging all 29 functional connectivity matrices for each patient and every frequency band. A two-sample two-tailed *t*-test was conducted to determine the functional connection differences between responders and non-responders. Statistical significance was set at *p* < 0.05. Results were assessed without and with correction for multiple comparisons using the false discovery rate (FDR) [19].

### 2.6. Graph Theoretical Analysis

Weighted connectivity matrices were obtained by applying a series of thresholds to the 19 × 19 weighted adjacency matrix of each subject and frequency band. The thresholds were set to the 90th, 85th, 80th, …, and 10th percentiles of the weights in the matrix, resulting in 17 weighted adjacency matrices with densities of 10%, 15%, …, and 90%. The weighted connectivity matrices were analyzed using indexes based on the graph theory [20]. In graph theoretical analysis, the brain is modeled as a graph composed of nodes, which indicate EEG channels, and undirected edges, which indicate functional connections determined by PLV. Graph-based parameters were estimated using MATLAB functions collected in the Brain Connectivity Toolbox (https://sites.google.com/site/bctnet/, 24 June 2021). For each of the reconstructed graphs, the following indices were estimated [21]: (1) basic measures (nodal strength), (2) measures of integration (global efficiency), (3) measures of segregation (clustering coefficient), and (4) measures of centrality (betweenness centrality).

### 2.7. Statistical Analysis

The comparison of patients’ data, divided in two groups, was carried out by Mann–Whitney U tests (age, number and dosage of AEDs, seizure frequency, blood sugar, and βHB) and Chi-squared tests (gender, seizure types, etiology of epilepsy, EEG lateralization, and dominant hand).

Graphs of the theoretic properties of responders and non-responders were compared using the two-sided Wilcoxon rank sum test before KDT and after six months of KDT. Comparing network properties before and after six months of KDT required the two-sided Wilcoxon signed-rank test to assess responders and non-responders. Statistical significance was set at *p* < 0.05, corrected for multiple comparison by false discovery rate (FDR). Subsequently, we performed a linear regression analysis between graph indices in which we identified statistically significant group differences and seizure reduction rates. The seizure reduction rate throughout the six months of assessment was determined, as follows: (baseline seizure frequency—seizure frequency in six months)/baseline seizure frequency ×100%.

## 3. Results

### 3.1. Patient Enrollment

As shown in Appendix A, we enrolled 46 subjects initially. Ten of them dropped out before the 3-month follow-up, and one before the 6-month follow-up due to children’s factors, parents’ factors, poor efficacy, or death. Four subjects were excluded for incomplete EEG data. Thirty-one subjects were finally included in this study, of whom 17 were KDT responders, and 14 were non-responders.

### 3.2. Patient Demographics and Response to KDT

The patients’ demographic data are shown in Table 1 and Appendix A. The mean age (age range) of our subjects was 6.69 ± 5.58 years (3 months−19 years and 1 month) in the entire group of 31 subjects, 6.7 ± 1.4 years (9 months−5 years and 4 months) in the responder group, and 7.33 ± 6.09 years (3 months−19 years and 1 month) in the non-responder group. Eight (47.1%) responders and seven (50.0%) non-responders were male. Most subjects over 2 years old were right-handed, and the EEG revealed bilateral-hemispheric epileptic discharges in most subjects. Eight (47.1%) responders and eight (57.1%) non-responders had the main seizure type of generalized onset motor seizure; others had focal onset seizures or multiple seizure types. Seizure etiologies of patients included genetic causes, hypoxic–ischemic encephalopathy, structural insults, immune disease, or was unknown. The mean baseline number of anti-epileptic drugs was 2.76 ± 1.03 in the responder group and 2.86 ± 1.23 in the non-responder group. The mean baseline seizure frequency per month was 103.59 ± 137.69 times in the responder group and 63.64 ± 77.20 times in the non-responder group. We identified no statistical difference between responders and non-responders in any of their baseline conditions, i.e., age, gender, EEG lateralization, dominant hand, main seizure type, seizure etiology, numbers and dosage of AEDs, or in baseline seizure frequency.

After receiving six months of KDT, the average βHB levels were 2.54 ± 1.82 mmol/L in responders and 2.78 ± 1.86 mmol/L in non-responders, without a statistical difference (*p* = 0.836). The mean seizure frequencies in responders and non-responders were significantly different, as expected (20.82 ± 39.23 versus 59.36 ± 70.87 times per month).

### 3.3. Strengths of Connections between Nodes

Before KDT, the connection strength in responders decreased throughout alpha and beta frequency bands compared with that of non-responders (Figure 1A). In responders, the connection strength decreased in delta, alpha, and beta frequency bands after six months of KDT compared to values determined prior to KDT (Figure 1B). However, in non-responders, increased connection strengths were observed in theta, alpha, and beta frequency bands after six months of KDT versus values determined prior to KDT (Figure 1C). After six months, responders exhibited decreased connection strength in the left hemisphere for delta, theta, alpha, and beta frequency bands compared with non-responders (Figure 1D). These results were not corrected for multiple comparisons with FDR.

### 3.4. Functional Connectivity

There were significant differences in network properties of responders and non-responders to KDT. In contrast, global efficiency, the mean clustering coefficient, mean nodal strength, and mean betweenness centrality in delta, theta, alpha, and beta frequency bands were not significantly different across proportional thresholds between responders and non-responders before KDT (Figure 2). Significant increases in network properties of mean betweenness centrality in the beta frequency band after six months of KDT were observed in responders to treatment (Figure 3). Non-responders displayed significant increases in network properties of the mean clustering coefficient and mean nodal strength in the theta frequency band after six months of KDT (Figure 4). After six months of KDT, responders exhibited a significant decrease in global efficiency, mean clustering coefficient, and mean nodal strength in the beta frequency band for a consecutive range of weighted proportional thresholds, compared with non-responders (Figure 5). There was an increase in mean betweenness centrality in theta, alpha, and beta frequency bands for responders after six months of KDT versus prior to treatment (Figure 5).

### 3.5. Correlations with Clinical Variates

In our study, no significant correlation was observed between βHB levels and seizure reduction rates. The alteration of mean betweenness centrality was significantly and positively correlated with seizure reduction rates in alpha (*R* = 0.453, *p* = 0.0105), beta (*R* = 0.453, *p* = 0.0016), and theta (*R* = 0.381, *p* = 0.0347) frequency bands in weighted adjacency matrices with densities of 90% (Figure 6). There were no significant relationships identified between seizure reduction rate and other graphic indices (global efficiency, mean clustering coefficient, and mean nodal strength). The alteration of mean betweenness centrality was significantly and negatively correlated with βHB levels in alpha (*R* = −0.522, *p* = 0.0062) frequency bands in weighted adjacency matrices with densities of 90%. There was no relationship between bloods sugar and alterations in mean betweenness centrality.

## 4. Discussion

Interictal EEG activity has been discussed in many studies, and emerging evidence has shown its association with seizure occurrence. Interictal epileptiform discharges, pathological high-frequency oscillation (80–200 Hz), pathological synchrony, focal EEG slowing, and hypsarrhythmia were all associated with poorly controlled seizures in the epileptic brain [22]. In this study, we used graph theory-based connectivity as a more precise method to measure subtle changes of its activity induced by KDTs. Our results provide evidence that the strengths of interictal connections between nodes became weaker in responders to therapy after six months of KDT compared to that before KDT. In contrast, stronger connectivity was noticed in non-responders to therapy.

### 4.1. Neuronal Synchrony Was Persistently High in Non-Responders after Six Months of KDTs

In the sequential EEG of KD non-responders, we found that clustering co-efficiency and nodal strength of the theta band increased after six months of KDT compared to values before KDT. Increased local neuronal synchrony within the epileptic brain has been discussed in previous studies using several quantitative measurements, including spectral coherence [23], magnitude-squared coherence [24], and mean phase coherence [25]. In our study of graph theory, clustering co-efficiency and nodal strength may also serve as measures of synchrony.

Although theta waves generated from the hippocampus are known to have an anticonvulsive effect [26], increased theta band connectivity was found to be related to a greater number of seizures in patients with brain tumors [27]. In our study, increased theta band connection seemed to be a predisposing factor for a poor KD response.

### 4.2. Global Connectivity Was Weaker in Responders Than in Non-Responders after Six Months of KDTs

When comparing responders and non-responders, connectivity at six months of KDT, especially in the beta band, significantly differed for global efficiency, mean clustering co-efficiency, and mean nodal strength, and these parameters were all weaker in responders. Most of these differences appeared to have a proportional threshold greater than 0.4, indicating that the major effect of KDTs on epilepsy occurred in low- and moderate-intensity connections. In agreement with our results, by the application of resting-state functional magnetic resonance imaging (MRI), Jie Song et al. found that global efficiency increased in epileptic patients compared to healthy people. They illustrated that an increased number of weak functional connections existed between networks of epileptic patients [28].

### 4.3. Betweenness Centrality Intensified in Responders after Six Months of KDTs

After six months of KDT, we observed a significant increment in mean betweenness centrality in the beta frequency band in responders to treatment as compared with that before KDT. While most connectivity appeared to be weaker in responders after six months of KDT, betweenness centrality strengthened. Betweenness centrality might represent localized connectivity and is presumed to play a protective role in seizure outspread or recurrence. Therefore, KDTs strengthens localized connectivity and leads to successful seizure control.

In studies of functional MRI, Jie Song et al. determined that significantly greater localized efficiency occurred in healthy controls compared to epilepsy patients [28]. In patients with mesial temporal lobe and focal neocortical epilepsy, increased connectivity was found in the epileptogenic zone, and decreased connectivity was found in distal networks. However, it is still unclear whether reductions in global connectivity are associated with cortical dysfunction from recurrent seizures or if they are a protective factor that prevents seizure outspread from the epileptogenic zone [29].

### 4.4. Betweenness Centrality Changes were Positively Correlated with Seizure Reduction Rates

Our correlation plot revealed a positive correlation between mean betweenness centrality changes and seizure reduction rates. This result shows the potential of using betweenness centrality as a parameter to monitor the response to KDTs. This is important not only in KDTs, but also in the field of epilepsy surgery, where Petroula Laiou et al. found that nodes with high betweenness centrality are possible targeting hubs for surgical resection [30].

Moreover, we found a significantly negative correlation between βHB levels and the alteration of mean betweenness centrality in alpha frequency bands. In our study, the βHB levels had a non-significant correlation with seizure reduction rates, possibly because of the small number of subjects. The reason for this unexpected finding and the underlying molecular mechanism is still an unsolved issue. Previous studies revealed positive correlations between seizure occurrence, delta band power, and allopregnanolone levels [31]. In addition, reduced plasma ghrelin levels are correlated to the maintenance of KTDs in epileptic children [32]. These evidence indicates that some plasma molecules may be correlated to brain connectivity and influence seizure control. How plasma molecules affect on connectivity is worthy to be investigated.

### 4.5. Future Prospective

Graph-theoretical functional connectivity has been used recently to analyze outcomes of other epileptic management strategies. Carboni et al. found that reduced interictal network connectivity of global, hemispheric, and lobar efficiency occurred in patients with good outcomes after epileptic surgery [33]. Jin Zhu et al. revealed changes in nodal efficiency, degree centrality, nodal local efficiency, and nodal shortest path length in different cerebral areas following vagus nerve stimulation surgery [34]. We have presented a pilot study that investigated graph theoretical brain connectivity during KDTs. Although the difference between responders and non-responders was not significant prior to KDTs, making us unable to predict its effect before the start of the therapy, several characteristics of connectivity alteration were revealed. In the future, directional/dynamic analysis, high-density (128-channel EEG) analysis, or analysis of functional MRI data may be performed to achieve more comprehensive results.

In addition, it is a challenging issue to maintain KDTs during the ongoing COVID-19 pandemic, and the use of telemedicine [35], e-health applications [36], and E-mail [37] can help the dietary management. We will develop a Taiwanese KetoApp, which, through the application of connectivity alteration at six months of KDTs, will help us choose the appropriate candidates for the KD program more precisely and facilitate the continuation of KD management.

### 4.6. Limitations

There are some limitations in our study. First, since refractory epilepsy accounts for only 20–30% of pediatric epilepsy and patients eligible for KDTs were even less, the study population was relatively small. As we designed a strict protocol of KDTs with a multi-professional team, standardized macronutrient ratio, precise laboratory test, detailed record of seizure frequency and pattern, and serial video EEG follow-ups, a multicenter study is challenging to carry out and is one of our future goals. Second, the background noise was difficult to filter completely by manipulation. While the performance of the denoise and artifact removal function is still limited, we examined the EEG to guarantee that muscle movement, head motion, or channels with poor signal were not involved and selected EEG sections with relatively good quality for further processing and analysis. Finally, in healthy subjects, we demonstrated the four sleep stages by examining background activity and some characteristic waves. As these features disappeared in patients with severe epileptic encephalopathy, it was difficult to define the sleep stages clearly.

## 5. Conclusions

KDTs tended to modify minor-to-moderate-intensity connections in patients’ brain. The reduction of global connectivity and the increment of betweenness centrality after six months of KDTs were associated with better responsiveness to it.

## Figures and Tables

**Figure 1 nutrients-13-02186-f001:**
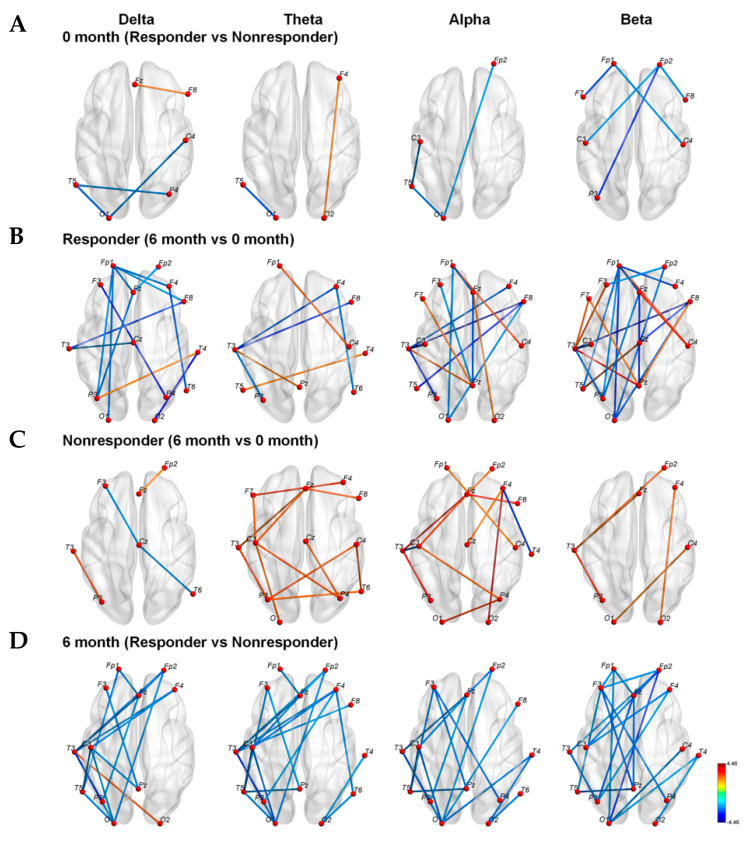
Network-based statistics between responders and non-responders or between before KDT and after six months of KDT in the delta, theta, alpha, and beta frequency bands; (**A**) The comparison of networks between responders and non-responders before starting KDT; (**B**) The networks of responders after 6 months’ KDT compared to that before KDT; (**C**) The comparison of 6 months to before usage in non-responders; (**D**) The responders’ networks compared to non-responders’ after 6 months of KDT. (KDT: ketogenic diet therapy.)

**Figure 2 nutrients-13-02186-f002:**
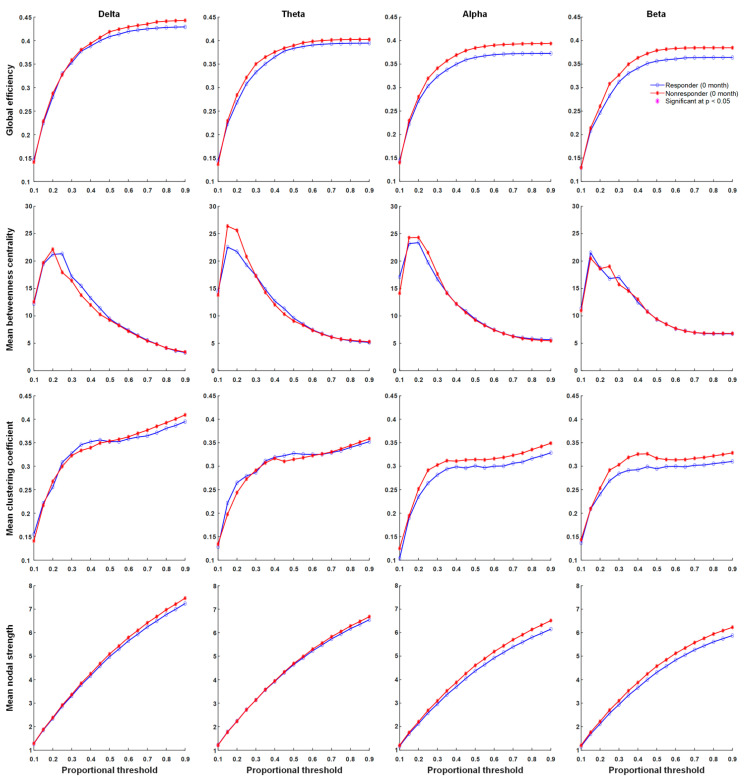
Graph theory-based analyses between responders and non-responders before KDT show global efficiency, mean clustering coefficient, mean nodal strength, and mean betweenness centrality in delta, theta, alpha, and beta frequency bands. Responders had decreased rather than increased connection strength in delta, alpha, and beta frequency bands compared with values before KDT. Asterisks denote statistically significant differences (*p* < 0.05). (KDT: ketogenic diet therapy.)

**Figure 3 nutrients-13-02186-f003:**
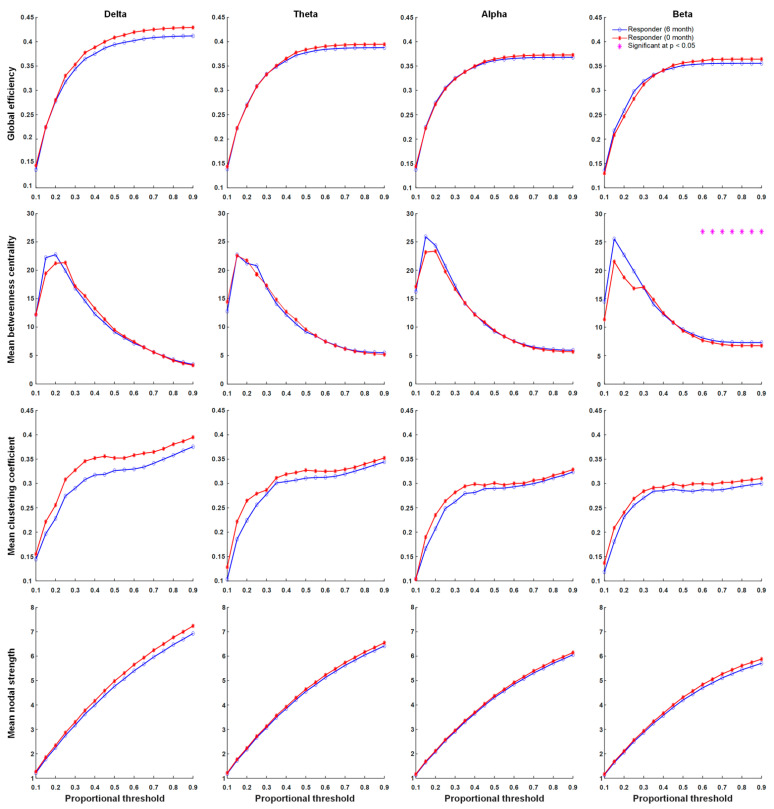
Graph theory-based analyses between before KDT and after six months of KDT for responders show global efficiency, mean clustering coefficient, mean nodal strength, and mean betweenness centrality in delta, theta, alpha, and beta frequency bands. The results revealed significant increases in the network properties of mean betweenness centrality in the beta frequency band after six months of KDT. Asterisks denote statistically significant differences (* *p* < 0.05). (KDT: ketogenic diet therapy.)

**Figure 4 nutrients-13-02186-f004:**
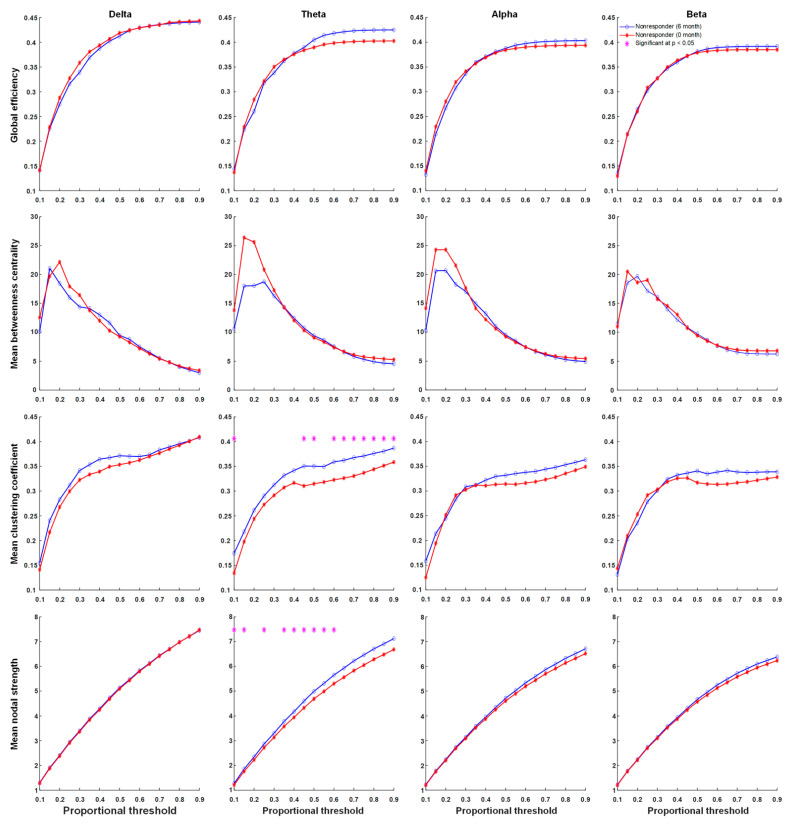
Graph theory-based analyses between before KDT and after six months of KDT for non-responders show global efficiency, mean clustering coefficient, mean nodal strength, and mean betweenness centrality in delta, theta, alpha, and beta frequency bands. The results revealed significant increases in the network properties of mean clustering coefficient and mean nodal strength in the theta frequency band after six months of KDT. Asterisks denote statistically significant differences (* *p* < 0.05). (KDT: ketogenic diet therapy.)

**Figure 5 nutrients-13-02186-f005:**
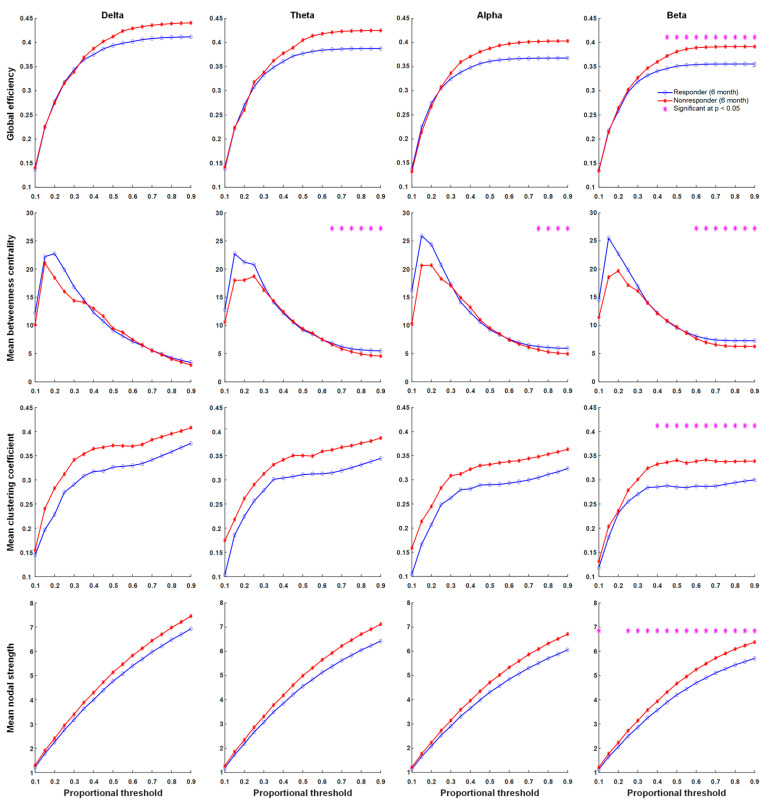
Graph theory-based analyses between responders and non-responders to KDT after six months of KDT show global efficiency, mean clustering coefficient, mean nodal strength, and mean betweenness centrality in delta, theta, alpha, and beta frequency bands. Responders exhibited a significantly decrease in global efficiency, mean clustering coefficient, and mean nodal strength in the beta frequency band for a consecutive range of weighted proportional thresholds. The mean betweenness centrality was increased in the theta, alpha, and beta frequency bands for responders after six months of KDT. Asterisks denote statistically significant differences (* *p* < 0.05). (KDT: ketogenic diet therapy.)

**Figure 6 nutrients-13-02186-f006:**
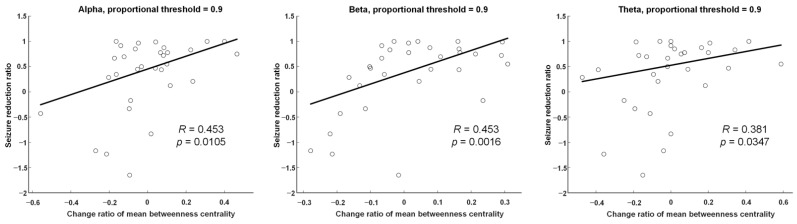
Linear regression analysis was used to explore the relationship between alteration of mean betweenness centrality of weighted adjacency matrices with densities of 90% and seizure reduction rate in alpha (*R* = 0.453, *p* = 0.0105), beta (*R* = 0.453, *p* = 0.0016), and theta (*R* = 0.381, *p* = 0.0347) frequency bands.

**Table 1 nutrients-13-02186-t001:** Patient Demographics. Statistical comparison of clinical data between responders and non-responders. The 6M seizure frequency means the average frequency during the first 6 months of ketogenic diet therapy. Responders were those who achieved a seizure reduction rate of 50% or more, showing a significantly lower 6M seizure frequency compared to baseline. No statistical difference was identified between the two groups in the baseline condition, in the biochemical indicators during ketogenic diet therapy, or in brain lateralization. The dominant hand is not applicable in subjects younger than 2 years old.

	Responders	Non-Responders	*p*-Value
Age (years, Mean ± SD)	6.69 ± 5.58	7.33 ± 6.09	0.409
Gender (n, %)			0.870
Male	8 (47.1)	7 (50.0)	
Female	9 (52.9)	7 (50.0)	
Main Seizure type (n, %)			0.816
Focal onset	8 (47.1)	5 (35.7)	
Generalized onset	8 (47.1)	8 (57.1)	
Both	1 (5.9)	1 (7.1)	
Etiology			0.395
Structural	1 (5.9)	2 (14.3)	
Genetic	7 (41.2)	7 (50.0)	
Infectious	0 (0.0)	1 (7.1)	
Metabolic	0 (0.0)	0 (0.0)	
Immune	0 (0.0)	1 (7.1)	
HIE	4 (23.5)	1 (7.1)	
Unknown	5 (29.4)	2 (14.3)	
No. of AEDs (Mean ± SD)	2.76 ± 1.03	2.86 ± 1.23	0.917
Dosage of AEDs (mg/kg/day, Mean ± SD)			
LEV	30.21 ± 14.48	30.32 ± 19.19	0.934
VPA	26.34 ± 12.95	23.90 ± 11.92	0.462
VGB	65.54 ± 40.88	86.79 ± 24.71	0.602
CZP	0.05 ± 0.03	0.04 ± 0.02	1.000
TPM	4.55 ± 2.50	2.96 ± 1.25	0.180
LTG	2.38	1.19 ± 0.84	0.157
PB	5.40 ± 0.89	4.14 ± 1.73	0.480
OXC	23.62 ± 8.52	16.51 ± 6.35	0.355
PER	0.02	0.53	0.317
Baseline seizure frequency (per month, Mean ± SD)	103.59 ± 137.69	63.64 ± 77.20	0.905
6M seizure frequency (per month, Mean ± SD)	20.82 ± 39.23	59.36 ± 70.87	0.011 *
6M Fasting blood sugar (mg/dl)	74.63 ± 11.31	77.46 ± 12.64	0.455
6M βHB (mmol/L)	2.54 ± 1.82	2.78 ± 1.86	0.836
EEG lateralization (n, %)			0.748
Right	2 (11.8)	3 (21.4)	
Left	1 (5.9)	1 (7.1)	
Bilateral	14 (82.4)	10 (71.4)	
Dominant hand (n, %)			0.493
Right	13 (76.5)	8 (57.1)	
Left	1 (5.9)	1 (7.1)	
N/A (<2 years)	3 (17.6)	5 (35.7)	

HIE: hypoxic ischemic encephalopathy; AED: antiepileptic drug; LEV: Levetiracetam; VPA: Valproate; VGB: Vigabatrin; CZP: Clonazepam; TPM: Topiramate; LTG: Lamotrigine; PB: Phenobarbital; CLB: Clobazam; OXC: Oxcarbazepine; PER: Perampanel; 6M: after 6 months of ketogenic diet therapy; βHB: beta-hydroxybutyrate; EEG: electroencephalography; N/A: not applicable; * *p* < 0.05.

## Data Availability

Not applicable.

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
