# Peer review of "Graph Theory-Based Electroencephalographic Connectivity and Its Association with Ketogenic Diet Effectiveness in Epileptic Children"

_nutrients, 2021, doi:10.3390/nu13072186_

Round 1

Reviewer 1 Report

1) The aim of the following manuscript “Graph Theory-based Electroencephalographic Connectivity in Epileptic Children Receiving Ketogenic Diet Therapy” is stated clearly, and the results are of interest. However, in my opinion, the discussion could be improved. Indeed, advancement in understanding in mechanism of epilepsy and dietary treatment should be considered as an important aspect. For this reason, mechanisms behind changes in ectroencephalographic connectivity in epileptic children receiving ketogenic diet should be deeply discussed by authors. For instance, it was recently published that allopregnanolone hippocampal levels were positively related to the seizure occurrence and to the power of delta band after the administration of an antiepileptic drug in an animal model of temporal lobe epilepsy (Costa et al., 2021). Do you think electroencephalographic connectivity in epileptic children receiving ketogenic diet therapy could be related to specific molecular changes?

Reference: Costa, A.-M.; Lucchi, C.; Malkoç, A.; Rustichelli, C.; Biagini, G. Relationship between Delta Rhythm, Seizure Occurrence and Allopregnanolone Hippocampal Levels in Epileptic Rats Exposed to the Rebound Effect. Pharmaceuticals 2021, 14, 127. https://doi.org/ 10.3390/ph14020127

2) Moreover, the activation of alimentary pattern generators and hormonal changes may exert modulatory effect on seizure occurrence. For instance, it has been demonstrated that ghrelin plasma levels were consistently reduced in children with refractory epilepsy and maintained on the KD (Marchiò et al., 2019). Do you think a relationship could exist between the observed electroencephalographic changes, and ghrelin plasma levels reduced in children with refractory epilepsy and maintained on the KD?

Reference: Marchiò, M.; Roli, L.; Lucchi, C.; Costa, A.M.; Borghi, M.; Iughetti, L.; Trenti, T.; Guerra, A.; Biagini, G. Ghrelin Plasma Levels After 1 Year of Ketogenic Diet in Children with Refractory Epilepsy. Front. Nutr. 2019, 6, 112.

3) It was demonstrated that a correct management of the ketogenic diet in pediatric patients with drug resistant epilepsy is important from the beginning to avoid side effects. Thus, the initiation and maintenance of the treatment are the result of concomitant efforts of pediatric neurologists, dieticians, families and other caregivers. Then, it has been also suggested that different kinds of e-health applications should be used simultaneously, as complementary resources, to improve epileptic patient outcomes in the management of the ketogenic diet (Costa et al., 2021). In this case, do you think the use of e-health technologies, together with the monitoring of the changes in the ectroencephalographic connectivity could increase patient outcomes?

Reference: Costa, A.-M.; Marchiò, M.; Bruni, G.; Bernabei, S.M.; Cavalieri, S.; Bondi, M.; Biagini, G. Evaluation of E-Health Applications for Paediatric Patients with Refractory Epilepsy and Maintained on Ketogenic Diet. Nutrients 2021, 13, 1240. https:// doi.org/10.3390/nu13041240

Author Response

Thank you very much.

Reviewer 2 Report

In a nutritional study, participants can always be grouped as responders and non-responders, so what is the purpose of this study? There is no control group which would include participants that were not on Ketogenic diet therapy. Thus, in the absence of this group it not clear what is the relevance of this study. 

It would be useful to state a hypothesis and address the significance of this study.

  1. The authors themselves provided several limitations especially the sample size of this study.
  2. There is no knowledge gap addressed in this article. It is study that groups a small group of children/teens suffering from epilepsy into responders vs non-responders. 
  3. It is not clear, what was the goal of this study? What was the hypothesis and the final conclusion?

Author Response

Thank you very much.

Round 2

Reviewer 1 Report

Accept 

Author Response

Thank you very much.

Reviewer 2 Report

This version is more clear and improved.

Author Response

Thank you very much.